

# Squirt flow due to interfacial water films in hydrate bearing sediments

Kathleen Sell[1†], Beatriz Quintal[2], Michael Kersten[1] & Erik H. Saenger[3,4]

[1] Johannes Gutenberg-University Mainz, Germany

[2] University of Lausanne, Switzerland

[3] International Geothermal Center, University of Applied Sciences Bochum, Germany

[4] Ruhr University Bochum, Germany

[†] Corresponding author (sell@uni-mainz.de)

**ABSTRACT**

Sediments containing gas hydrate dispersed in the pore space are known to show a characteristic seismic anomaly which is a high attenuation along with increasing seismic velocities. Currently, this observation cannot be fully explained albeit squirt-flow type mechanisms at the microscale have been speculated to be the cause. Recent major findings from in-situ experiments coupled with high-resolution synchrotron-based X-ray micro-tomography revealed a systematic presence of thin water films between the quartz grains and the encrusting hydrate. In this study, the data was obtained from the experiments and underwent an image processing procedure to quantify the thicknesses and geometries of the aforementioned interfacial water films. Overall, the water films vary from sub-μm to a few μm in thickness where some of them are interconnected by water bridges. This geometrical analysis is then used to propose a new conceptual squirt flow model for hydrate bearing sediments. Subsequently the established model acts as a direct model input to obtain seismic attenuation. Our results support previous speculations that squirt flow can explain high attenuation at seismic frequencies in hydrate bearing sediments, but based on a conceptual squirt flow model which is different than those previously considered.

Keywords: attenuation, squirt flow, interfacial films, dispersion, micro-tomography, gas hydrates, sediments, numerical modeling

## 1. INTRODUCTION

Important mechanisms of wave attenuation in fluid-saturated porous media from seismic to ultrasonic frequencies, include friction between grain boundaries (Winkler and Nur, 1982), global flow or Biot's mechanism (Biot, 1962), and wave-induced fluid flow at mesoscopic and microscopic scales (e.g., Müller et al., 2010). At the mesoscopic scale, patchy saturation and fractures are the most prominent causes of wave-induced fluid flow (White, 1975; White et al.,



1975; Brajanovski et al., 2005; Tisato and Quintal, 2013; Quintal et al., 2014). At the microscopic scale, wave-induced fluid flow is commonly referred to as squirt flow and typically occurs between interconnected microcracks or between grain contacts and stiffer pores (O'Connell and Budiansky, 1977; Murphy et al., 1986; Mavko and Jizba, 1991; Sams et al., 1997; Adelinet et al., 2010; Gurevich et al., 2010). The attenuation caused by global flow as well as that caused by wave-induced fluid flow at microscopic or mesoscopic scales are frequency dependent, but while the latter can have a strong effect at seismic frequencies (Pimienta et al., 2015; Subramaniyan et al., 2015; Chapman et al., 2016), global flow will only cause significant attenuation at ultrasonic frequencies or higher (e.g., Bourbie et al., 1987). The attenuation caused by friction between grain boundaries is, on the other hand, frequency independent and basically depends on the confining pressure and the strain imposed by the propagating wave (Winkler and Nur, 1982). Its effect is expected to be small for the correspondingly small strains caused by seismic waves used in exploration and reservoir geophysics. Furthermore, the attenuation caused by wave-induced fluid flow tends to be linearly superposed to that due to friction between grain boundaries, as shown by Tisato and Quintal (2014).

Gas hydrates (GH) are ice-like structures comprised of gas molecules entrapped by water molecules (Sloan and Koh, 2008). The widespread global occurrence of GH and the fact that 1 $m^3$ of GH contains up to 164 $m^3$ of natural gas ($CH_4$ and $CO_2$ at standard conditions) draws attention to the idea of using GH as a potential future energy resource (Schicks et al., 2011). Nevertheless, GH-bearing sediments have been discussed not only as a relatively clean hydrocarbon reservoir (Collett and Ladd, 2000), but also in terms of a geohazard that can potentially contribute to global warming associated to hydrate dissociation and subsequent destabilization of GH-cemented deep sea sediments at continental margins (Kvenvolden, 1993; Nixon and Grozic, 2007). Occurrences of GH are restricted to locations providing the required amount of gas and water and the preferred pressure-temperature (p/T) conditions, which are commonly referred to as the so-called gas hydrate stability zones. Usually, GH reservoirs are mainly limited to marine continental margins, deep lakes and permafrost regions (Bohrmann and Torres, 2006).

In the search for GH reservoirs, the attenuation of seismic waves caused by the pore fluids might be an important survey tool (e.g. Bellefleur et al. 2007). However, little effort has been directed toward studying its effects for unconsolidated sediments hosting GH in a rather dispersed manner. GH forming in the pore space of unconsolidated sediments at given p/T-conditions alters the effective elastic and effective transport properties of the hosting sediment. It is known that the presence of GH in the sediment not only reduces the porosity and causes significant changes on its permeability, but also results in higher P- and S-wave velocities due to stiffening of the hosting matrix (Dvorkin et al., 2003; Guerin & Goldberg, 2005; Yun et al., 2005; Priest et al., 2006; Waite et al., 2009). In other words, the bulk and shear moduli increase due to the GH matrix-supporting effect within the sedimentary frame (Ecker et al., 1998). Additionally, the presence of GH causes higher attenuation of the seismic waves (Bellefleur et al. 2007; Dewangan et al. 2014) which was in particular observed for sediments containing dispersed GH in the pore space (Guerin and Goldberg, 2002; Dvorkin and Uden, 2004). This identified anomalous seismic behavior in terms of increased attenuation and velocities (Guerin



and Goldberg, 2002; Dvorkin and Uden, 2004) cannot be fully explained, although wave-
induced fluid flow at the microscopic and mesoscopic scales have has been speculated to cause
them (Priest et al., 2006; Gerner et al. 2007). Gerner et al. (2007) conducted numerical P-wave
velocity simulations in highly permeable sedimentary layers, similar to hydrate-bearing
sediments, and identified interlayer flow at the mesoscopic scale (White et al., 1975) as a
potential mechanism of attenuation. Other authors have considered classical squirt flow models
(O'Connell and Budiansky, 1977; Murphy et al., 1986) as the main source of attenuation in
hydrate-bearing sediments (Dvorkin and Uden, 2004; Guerin & Goldberg, 2005; Priest et al.,
2006; Waite et al., 2009; Marin-Moreno et al., 2017). Quantifying GH saturation levels through
geophysical exploration techniques is, however, not straightforward as there are still open
questions on GH formation, its microstructure and distribution in the natural settings.
Additionally, the recovery of unaltered natural GH samples is hampered due to their fast
decomposition under ambient conditions. Therefore, various researchers have attempted to
mimic the natural environment of GH-bearing sedimentary matrices in laboratory experiments
(Berge et al., 1999; Ecker et al., 2000; Dvorkin et al., 2003; Yun et al., 2005; Spangenberg and
Kulenkampff, 2006; Priest et al., 2006, 2009; Best et al., 2010, 2013; Hu et al., 2010; Li et al.,
2011; Zhang et al., 2011; Dai et al., 2012; Schicks et al., 2013). The results of this collective
effort established a number of conceptual models for the role of GH embedded in its
sedimentary matrix (Figure 1). Nevertheless, these approximations turned out to be still not
satisfactory. Although it has been suggested that all hydrate habits known from laboratory
investigation involving synthetic samples occur also in nature (Spangenberg et al. 2015), none
of those simplified models can yield accurate predictions of GH saturations from field electric
resistivity or seismic data alone (Waite et al., 2009; Dai et al., 2012).
In this study, we introduce an alternative conceptual model based on findings from in-situ
experiments coupled with high-resolution synchrotron-based X-ray micro-tomography
(Chaouachi et al., 2015; Sell et al., 2016). The 3D micro-tomography data for quartz sands
bearing GH revealed the presence of thin interfacial water films, between the pore-filling GH
and the grains, occasionally interconnected via water bridges, as well as water pockets
embedded in the GH. We perform numerical simulations of squirt flow in the proposed
conceptual model to study the related dispersion of the stiffness modulus and the corresponding
frequency-dependent attenuation. The results demonstrate the high levels of seismic
attenuation/dispersion that such features can cause and support the suggestions that the
estimation of GH saturation for GH occurring in a rather dispersed manner could be
accomplished by using P- and S-wave attenuation as a tool for indirect geophysical
quantification (Guerin and Goldberg, 2002; Priest et al. 2006; Best et al. 2013; Marin-Moreno
et al., 2017).



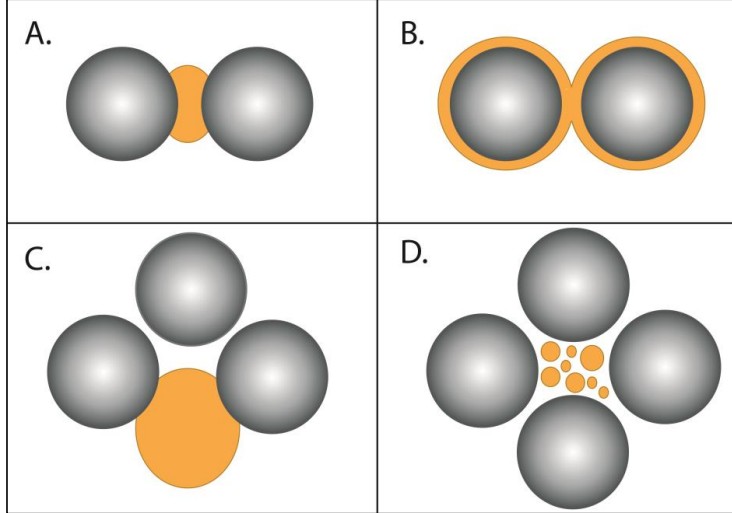

**Figure 1.** Review of the established conceptual models (Grains = grey and GH = orange), with (A)
cementation – GH cements the grains, (B) encrustation – GH coats the grains, (C) matrix-supporting –
GH is part of the sediment matrix, and (D) pore-filling – GH employs the pore space forming crystallites
of varying size (modified after Dai et al., 2004).
**2. THE INTERFACIAL WATER FILMS**
Chaouachi et al. (2015) conducted various in-situ experiments coupled with synchrotron-based
tomography at the TOMCAT beamline of the Paul Scherrer Institute. The aim was to study the
formation process and distribution of gas hydrates in various matrices, such as pure quartz sand
and glass beads, as well as mixtures of quartz sand with clay minerals. These in-situ
experiments have been realized using an experimental setup under elevated pressure and
lowered temperature. Further details are given by Chaouachi et al. (2015), Falenty et al. (2015),
and Sell et al. (2016).



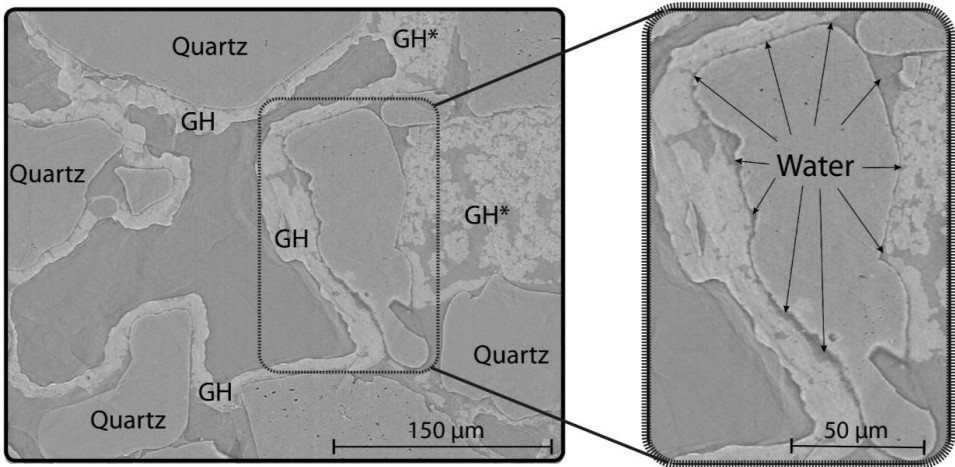

**Figure 2.** (Left) Overview of an unfiltered 2D slice in y,z-direction of quartz sand containing GH. Note that due to its unfiltered state, this image contains artifacts, such as streaks and slight edge enhancement. Phases can be identified on the base of grey scale differences.

In this study, the focus lies on samples containing pure natural quartz sand sieved at 200–300 µm grain size. Details on the sedimentology and mineralogy of the host sediment are provided by Chuvilin et al. (2011). We use a reconstruction process (Marone and Stampanoni, 2012) that yields an image matrix of $2560 \times 2560 \times 2160$ voxels, with an isometric voxel size of 0.74 and 0.38 µm at 10-fold and 20-fold optical magnification, respectively. The reconstructed tomograms revealed discernible grey value differences between the three relevant phases of the sample: solid grains, hydrate, and water (Figure 2). Image analysis has been accomplished to reduce image artifacts, such as inhomogeneity in grey scale values, streaks and edge enhancement by applying a systematic image enhancement workflow comprising different image filter combinations in 2D and 3D (Sell et al., 2016). One of the most interesting observations made was a systematic appearance of a thin interfacial water film separating the quartz grains from the GH phase (Chaouachi et al., 2015). This fluid interface was observed in samples where GH was formed in quartz sand samples directly from the juvenile state not involving GH dissociation, as well as where GH was formed from gas-enriched water.




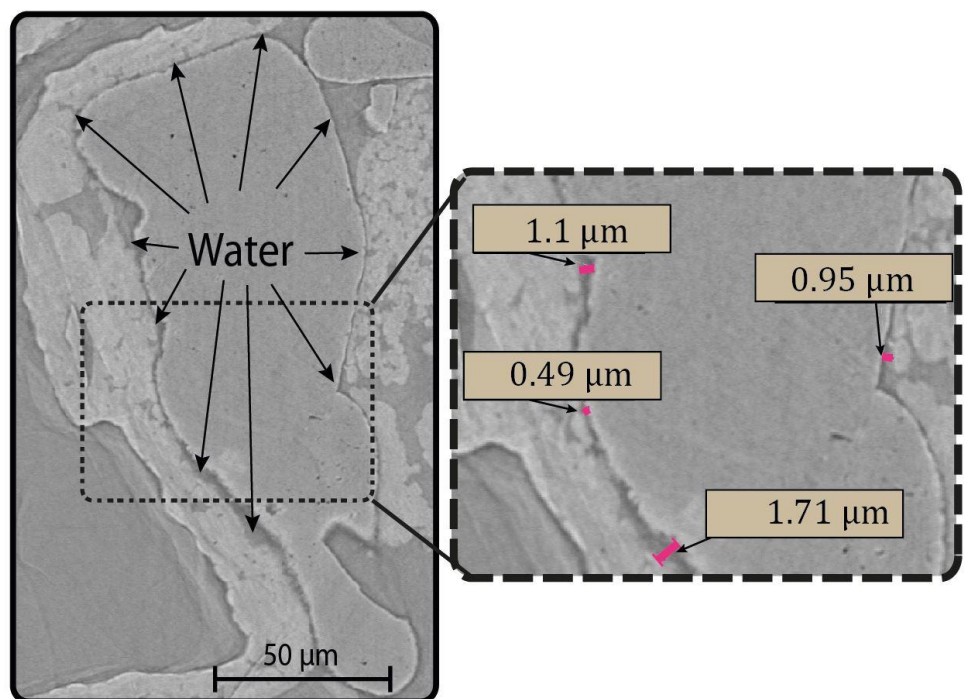

**Figure 3.** Raw (unfiltered) 2D image in y,z-direction at a spatial resolution of 0.38 m. The zoom depicts
the measurement of a thin interfacial water film varying in thickness from 0.49 ⬜m to 1.71 µm.
The broad range of grey scale values of the filtered images can be classified using watershed
segmentation combined with region growing tools of the software packages of Avizo Fire 7
(FEI, France) and Fiji. The full workflow has been described by Sell et al. (2016). Basically,
for this work the thickness variation and geometry of the water film has been determined
(Figure 3), an information needed to define our conceptual model to investigate on attenuation
in GH-bearing sedimentary matrices (Figure 6). The multi-phase model involves idealized
round-shaped grains covered by a homogenous thin water film and embedded in non-porous
or porous hydrate.

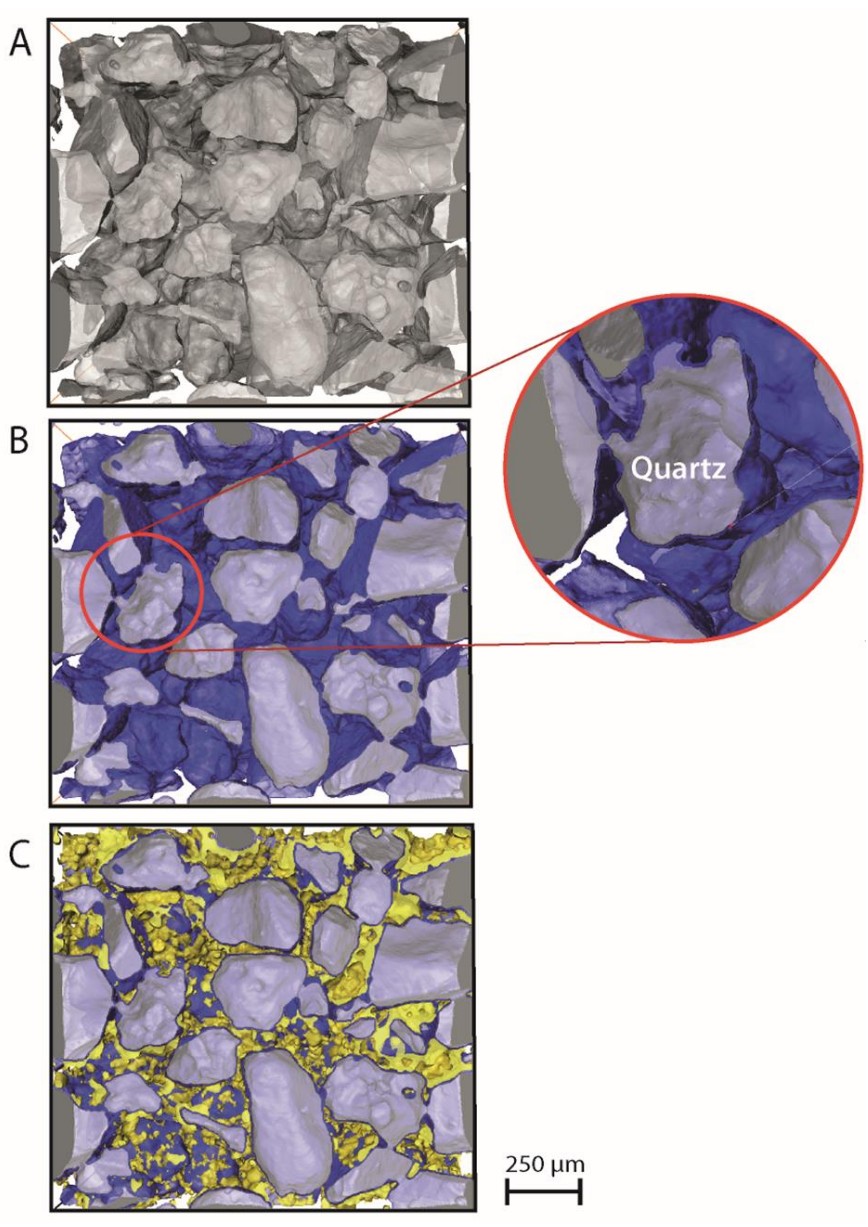

**Figure 4.** Volume-rendered phases in a representative image sample. For a better visualization, the
phases are introduced step-by-step, with (A) grains (grey), (B) grains and interfacial water films (blue),
and (C) grains, water film and hydrate (yellow). A zoom in (B) shows an interfacial water film measured
at $1 - 4$ voxels equivalent to $0.38 - 1.52$ µm thickness, respectively.




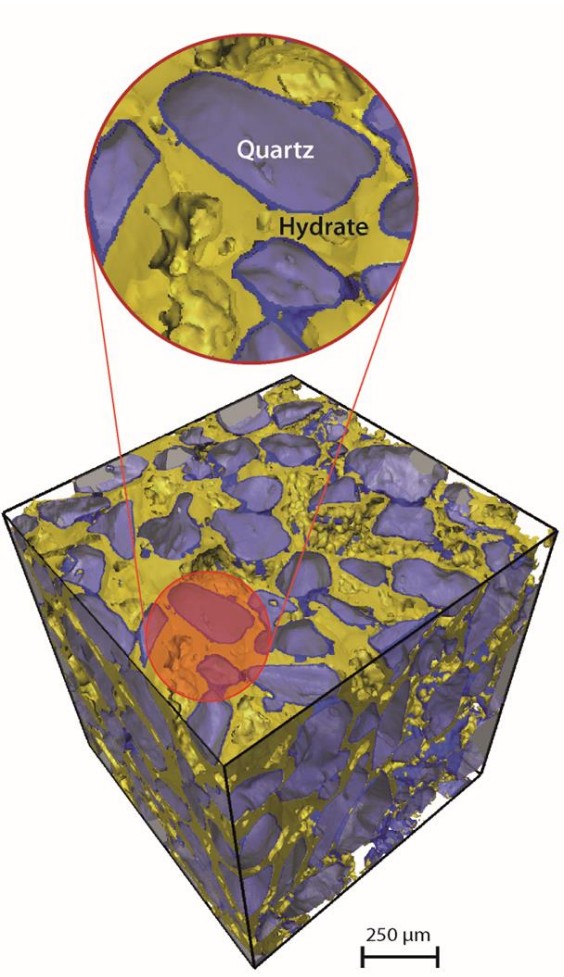

**Figure 5.** Volume-rendered image of a representative Region of interest (ROI) of 600 × 600 × 600
voxels at 0.38 μm spatial resolution. The zoom-in depicts quartz grains fully separated from the pore-
filling hydrate by thin interfacial water films, with two quartz grains having their water films
interconnected by a water bridge.



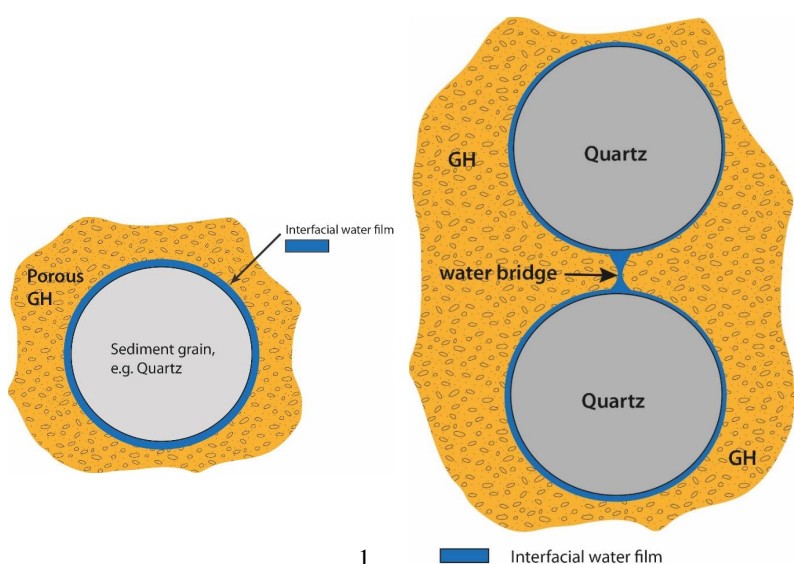

**Figure 6.** Schemes of (A) a new concept model for GH encrusting quartz grains separated by a thin interfacial water film and (B) connected by a water bridge.

## 3. NUMERICAL METHODOLOGY

### 3.1 Mathematical formulation

To estimate frequency-dependent attenuation in the GH systems described above we employ a hydromechanical approach (Quintal et al., 2016) based on the conservation of momentum

$$\nabla \cdot \boldsymbol{\sigma} = 0, \tag{1}$$

with the components $\sigma_{kl}$ of the stress tensor $\boldsymbol{\sigma}$ defined according to the general stress-strain relations in the frequency domain

$$\sigma_{kl} = 2\mu\varepsilon_{kl} + \left(K - \frac{2}{3}\mu\right)e\delta_{kl} + 2\eta\omega i\varepsilon_{kl} - \frac{2}{3}\eta\omega ie\delta_{kl}, \tag{2}$$

where $\varepsilon_{kl}$ denotes the components of the strain tensor, $e$ denotes the cubical dilatation given by the trace of the strain tensor, $\omega$ is the angular frequency, and $i$ represents the unit imaginary number. The indexes $k, l = 1, 2, 3$ refer to the three Cartesian directions $x_1, x_2, x_3$ or $x, y, z$ and $\delta_{kl}$ is the Kronecker delta ($\delta_{kl} = 1$ for $k = l$ and $\delta_{kl} = 0$ for $k \neq l$). The material parameters $\mu$, $K$, and $\eta$ are the shear modulus, the bulk modulus, and the shear viscosity, respectively.

Using this general mathematical formulation (equations 1 and 2), a heterogeneous medium can be described as having an isotropic, linear elastic solid frame and fluid-filled cavities or pores, to which a specific choice of material parameters can be assigned. The same unknowns and material parameters describe the behaviors of the solid and the fluid phases. For example, an unknown $u$ describes the solid displacement in the domains of the model representing an elastic



solid and also describes the fluid displacement in the domains representing a viscous fluid. In
fact, equation 2 reduces to Hooke's law by setting the shear viscosity to zero. In these regions,
$\mu$ and $K$ denote the shear and bulk moduli of the corresponding elastic solid, and the shear
viscosity $\eta$ is zero. In the model domains representing a compressible viscous fluid, the shear
modulus $\mu$ is set to zero and the combined equations 1 and 2 reduce to the quasi-static,
linearized Navier-Stokes' equations for the laminar flow of a Newtonian fluid (e.g., Jaeger et
al., 2007). In these fluid-filled regions, $K$ and $\eta$ denote the bulk modulus and shear viscosity of
the fluid.
When the aforementioned heterogeneous medium is deformed, fluid pressure differences
between neighbor regions induce fluid flow or, more accurately, fluid pressure diffusion, which
in turn results in energy loss caused by viscous dissipation (Quintal et al., 2016). At the
microscopic scale, this attenuation mechanism is commonly referred to as squirt flow (e.g.,
O'Connell and Budiansky, 1977; Murphy et al., 1986) and is the sole cause of attenuation in
our simulations, as we neglected the inertial terms in equations 1 and 2.
**3.2 Finite element modeling**
Our 2D problem is equivalent to a 3D case under plain strain conditions, which means no strain
outside the modeling plane is allowed to develop. For the corresponding simulations, we
consider the directions $x$ and $y$, to be in the modeling plane and direction z to be the one in
which no displacement or displacement gradients can occur.
The numerical solution is based on a finite-element approach in the frequency domain. We
employ an unstructured triangular mesh, which allows for an efficient discretization of slender
heterogeneities having large aspect ratios, such as the thin interfacial water films, by strongly
varying the sizes of the triangular elements (e.g., Quintal et al., 2014). A few elements across
the thin interfacial water film are necessary to accurately capture the viscous dissipation in this
region, while much larger elements are sufficient in the solid elastic domains. The sizes of
smallest and largest elements in our meshes differ by 3 orders of magnitude.
To assess the P-wave attenuation and modulus dispersion caused by squirt-flow, we subject a
rectangular numerical model to an oscillatory test. A sinusoidal downward displacement is
applied homogeneously at the top boundary of the numerical model. At the bottom, the
displacement in the ($y$) vertical direction is set to zero. At the lateral boundaries of the model,
the displacement in the ($x$) horizontal direction is set to zero. From this test, we obtain the stress
and strain fields, averaged over the entire model domain. The mean stress and strain are used
to compute the complex-valued and frequency-dependent P-wave modulus corresponding to a
wave propagating in the vertical direction. The real part of the P-wave modulus $H$ is used to
illustrate the P-wave modulus dispersion while the ratio between its imaginary and real parts
is used to quantify the P-wave attenuation $1/Q_P$. The S-wave attenuation and dispersion can be
evaluated in a similar manner simply by changing the boundary conditions to those of a simple-
shear test (e.g., Quintal et al., 2012, 2014).



Our 3D problem is solved similarly to the 2D problem using an unstructured mesh, but with
tetrahedral elements. Again, the element sizes in our 3D meshes vary by about 3 orders of
magnitude.
**4. NUMERICAL RESULTS**
Many sources of squirt flow might coexist in unconsolidated sediments hosting GH, such as
those resembling the conventional squirt flow models introduced by O'Connell and Budiansky
(1977) for interconnected microcracks and by Murphy et al. (1986) for microcracks or grain
contacts connected to spherical pores. Marin-Moreno et al. (2017) describes an integrated
approach that combines the effects of some squirt flow models and other attenuation
mechanisms. Here our objective diverges from that. We instead aim at studying the squirt flow
phenomenon and the resulting frequency-dependent attenuation associated with a specific
model, which is geometrically different from the mentioned conventional squirt flow models
and is based on the thin interfacial water films. We thus neglect all other potentials sources of
attenuation.
**4.1 Attenuation mechanism in a thin interfacial water film**
Our 2D numerical model domain corresponds to a fundamental block of a periodic distribution
of unconsolidated circular quartz grains dispersed in a continuous GH background and
separated from the latter by a thin interfacial water film (Figure 7). Aim of this basic model is
to have a first estimate of the possible attenuation effect by a thin interfacial water film. The
subdomain representing the thin interfacial water film is described by the corresponding
properties of this viscous fluid, while the other subdomains are described by properties of two
different elastic solids, quartz and GH. These properties are given in Table 1.
Based on the material properties given in Table 1, we consider thicknesses of the interfacial
water film ranging from 0.1 μm to 1 μm as well as two grain diameters 150 and 250 μm for the
2D model. These values were chosen considering the sizes of the quartz grains used in the
laboratory experiment from which the SRXCT data were obtained, which ranged from 150 to
300 μm, and the thicknesses of the interfacial water films observed in the data, ranging from
0.38 μm to 1.5 μm. Note that the thinnest interfacial water films observed were limited by the
highest achieved spatial resolution of 0.38 μm. Despite limitation of spatial resolution, the
water film thicknesses below 0.38 μm have been considered for our numerical analysis as well.
The numerical results are expressed as the real part of the P-wave modulus and the P-wave
attenuation $1/Q_P$ (Figure 8). We observe that a decrease in the thickness of the interfacial water
film causes the attenuation and dispersion curves to shift to lower frequencies. In fact, high
attenuation values $(1/Q \sim 0.1)$ are observed at seismic frequencies (~100 Hz) when the
interfacial water film is as thin as 0.1 μm and the grain diameter is as large as 250 μm.
Decreasing the grain diameter, on the other hand, causes a shift to higher frequencies of the
attenuation and dispersion curves.



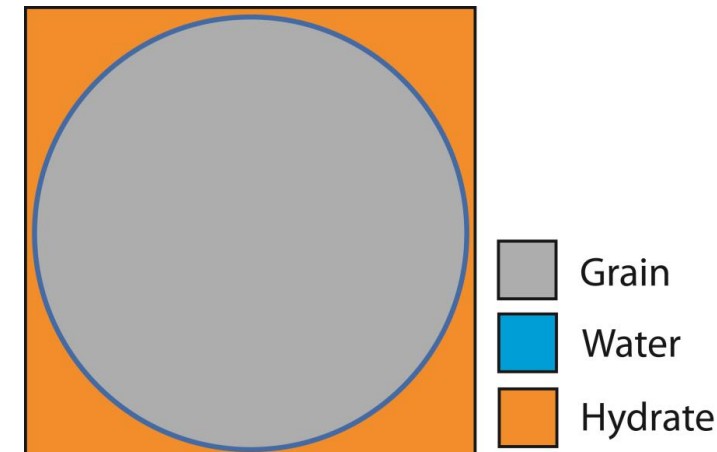

**Figure 7.** Fundamental block of an idealized periodic medium representing sediment grains which are
separated from the embedding GH background by a thin interfacial water film.
**Table 1.** Material properties used in the numerical simulations. *The properties of quartz are based on
the work of Bass (1995) and those of hydrate on Helgerud (2003).

| Material parameter | Quartz[*] | Hydrate[*] | Water |
|---|---|---|---|
| Shear modulus $\mu$ | 44.3 GPa | 13.57 GPa | 0 |
| Bulk modulus $K$ | 37.8 GPa | 8.76 GPa | 2.4 GPa |
| Shear viscosity $\eta$ | 0 | 0 | 0.003 Pa×s |





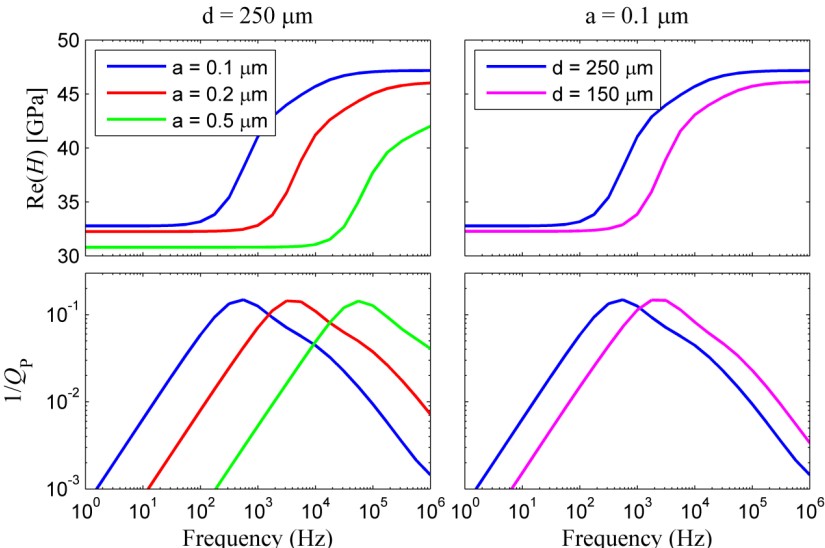

**Figure 8.** Real part of P-wave modulus, $H$, and corresponding P-wave attenuation, $1/Q_P$, as functions
of frequency, for the model shown in Figure 7, considering the grain diameter d and thickness a of the
interfacial water film, which are indicated in the legends and plot titles.

The geometry of the introduced model (Figure 7) is different than the classical squirt-flow
geometries involving interconnected plane cracks or a plane crack connected to a pore of low
aspect ratio. To better understand how dissipation occurs for this type of geometry, we initially
focus on the fluid pressure field $P$ (Figure 9) in the circular interfacial water film at the
characteristic frequency. The vertical compression of the model illustrated in Figure 7 causes
a larger deformation of the interfacial water film at the top and bottom of its circular geometry
than on the sides. This observation is comparable to horizontal cracks that are more deformed
by a vertical compression than vertical cracks in a classical squirt flow model. Here, the
heterogeneous deformation causes fluid pressure to increase. The most deformed parts which
are the top and the bottom, exhibit the highest fluid pressure, as shown in Figure 9. The pressure
gradient present in this heterogeneous pressure field induces fluid to be displaced from the
regions of higher pressure (top and bottom) towards the regions of lower pressure (sides).
Exemplarily, the components of the fluid velocity field in the $x$ and $y$ directions $V_x$ and $V_y$
(Figure 10) and its corresponding local attenuation field $1/q$ (Figure 11) are depicted in the
representative top-right quadrant of the model. Considering the symmetry of this process in the
four quadrants of the circular interfacial water film (Figure 9) it is reasonable to show only one
quadrant out of four.

In Figure 10 we observe the text-book (e.g., Jaeger et al., 2007) parabolic profile of the fluid
velocity across the interfacial water film, with larger fluid velocity in the center of the film,
governed by Navier-Stokes equations. This fluid velocity is associated with an energy
dissipation caused by viscous friction, shown in Figure 11. At the boundaries of the interfacial
water film, larger viscous friction explains the lower fluid velocity and larger energy



dissipation, in comparison to the center of the film. The attenuation is strongly reduced towards
the center of the film by a few orders of magnitude. Now looking at how these fields change
along the interfacial water film, we observe that the maximal velocity and attenuation (compare
Figures 10 and 11) coincide with the maximal pressure gradient (Figure 9). Whereas in the
middle of the higher pressure and lower pressure regions the pressure gradient is minimal
causing the fluid velocity and attenuation to drop drastically.

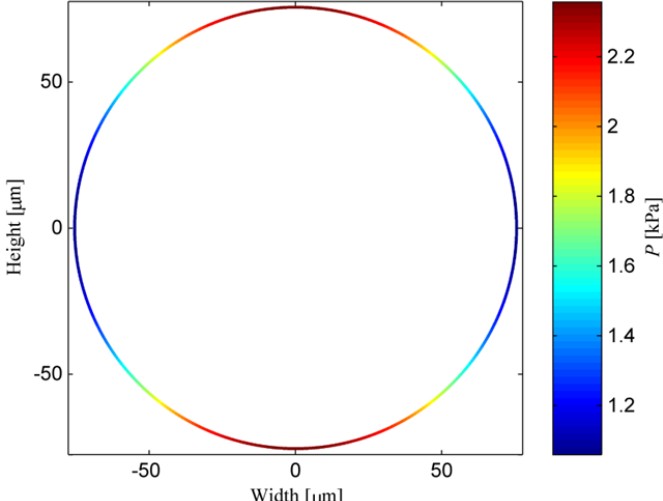

**Figure 9.** Fluid pressure $P$ for the model shown in Figure 7, considering a grain diameter d = 150 μm
and thickness of the interfacial water film a = 1 μm. The oscillation frequency is equal to the
characteristic frequency ($1.8 \times 10^6$ Hz).

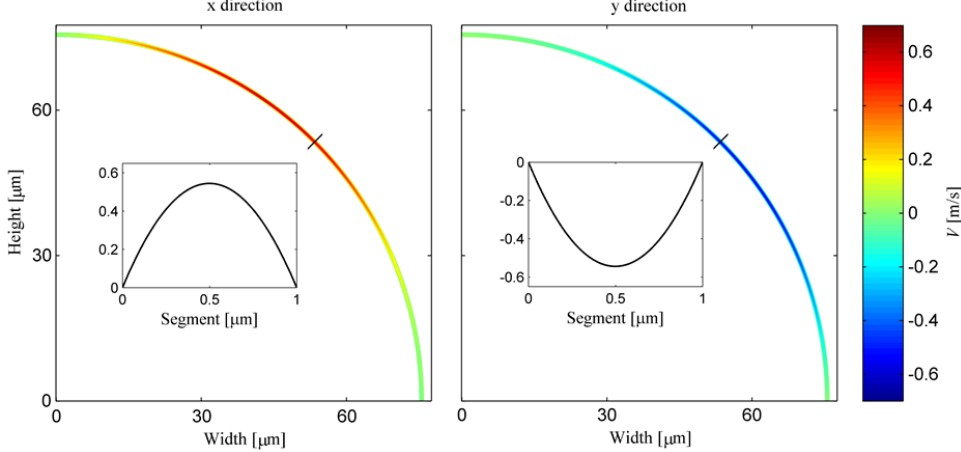

**Figure 10.** Zoom-in to the top-right quadrant of the model shown in Figure 9 showing the fluid velocity
components $V_x$ and $V_z$, for a grain diameter d = 150 μm, a thickness of the interfacial water film a = 1





μm, and at the characteristic frequency. These fields correspond to the fluid pressure field shown in
Figure 9. The insets illustrate the profiles across the interfacial film where it is crossed by a black line.

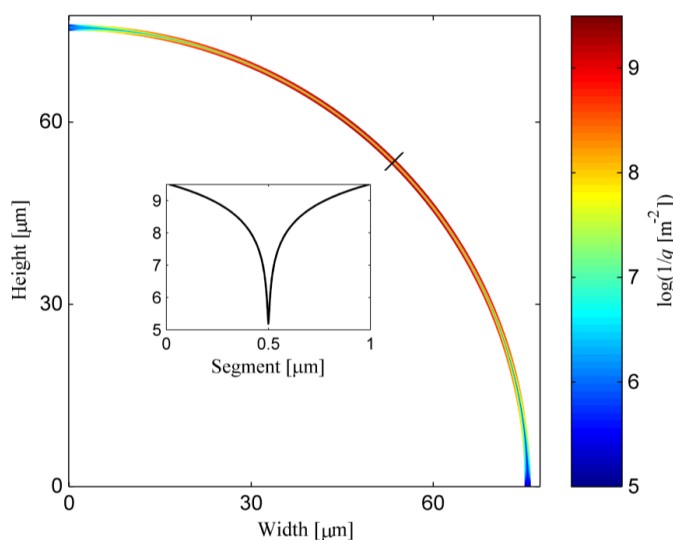

**Figure 11.** Zoom-in to the top-right quadrant of the model shown in Figure 7 showing the local
attenuation $1/q$, for a grain diameter d = 150 μm, with a water film thickness a = 1 μm, and at the
characteristic frequency. This field corresponds to those shown in Figures 9 and 10. The inset illustrates
the profile across the interfacial film where it is crossed by a black line.
**4.2 Effects of water pockets and water bridges**
In this subsection, a few alterations are added to the basic three-phase model illustrated in
Figure 7. These alterations are based on more detailed observations obtained from SRXCT
such as water pockets that have been detected in porous GH or a water bridge that might occur
connecting two neighboring interfacial water films (Figure 12). For this, the effect of these
features on the P-wave modulus dispersion and attenuation (Figure 13) is studied and compared
to results obtained from corresponding models where these features have not been considered.
The inclusion of water pockets has a modest effect on the attenuation and dispersion, while it
reduces the overall value of the P-wave modulus, as a certain volume of GH is replaced by a
much less stiff material (water). Concurrently, modest increase in attenuation is associated with
a more compressible effective background; no attenuation occurs within the water pockets.
The connecting water bridge introduces an additional length scale for the dissipation process,
as fluid flow and dissipation will also occur through this relatively short and wide path. This
explains the additional attenuation peak observed at higher frequencies, while the previous
peak at $2 \times 10^3$ Hz suffers a slight reduction in magnitude. A reduction in magnitude occurs
because the pressure equilibration process involving the water bridge causes a reduction in





pressure in the region connected to the bridge and thus a reduction of the previously discussed
(Figure 8) pressure gradient between this region and the sides of the circular interfacial water
film. The dispersion agrees with the attenuation curve with two inflections, corresponding to
the two attenuation peaks, between the high- and low-frequency limits.

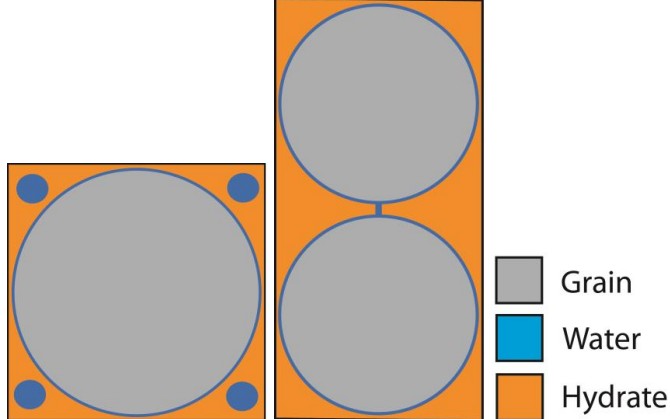

**Figure 12.** Fundamental blocks of two periodic media representing loose sandstone grains which are
separated from the embedding GH background by a thin interfacial water film. On the left water pockets
are located in the GH background and on the right the interfacial water films are connected to another
through a water bridge**.**





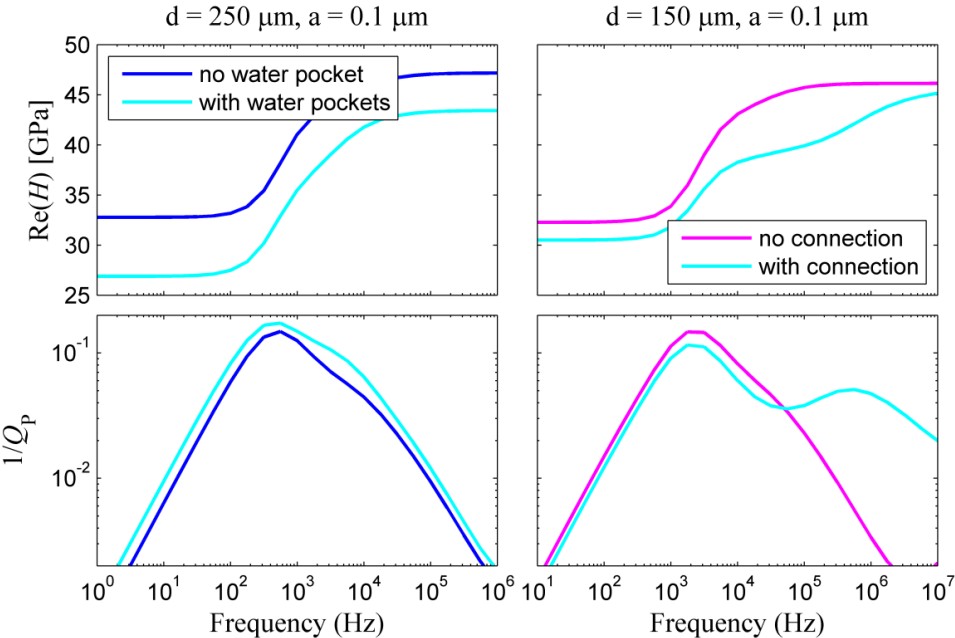

**Figure 13.** Real part of P-wave modulus, $H$, and corresponding P-wave attenuation, $1/Q_P$, as functions
of frequency, for the models shown in Figure 12 in comparison with the corresponding results from the
model shown in Figure 7 and given in Figure 8. The grain diameter d and thickness a of the interfacial
water film are indicated in the plot titles.

### 4.3 Evaluation of 3D effects

The following subsection considers a comparison between the results of the simulation
illustrated in Figures 9-11, for the 2D model shown in Figure 7, and those of a simulation
performed on its 3D counterpart. Our 3D model consists of a sphere in the middle of a cube
(Figure 14), consequently a centered cross section matches the 2D model shown in Figure 7.
The aperture of the water film is 1 μm and the grain diameter is 150 μm (as for Figures 9-11).
The numerical results are shown in Figure 15 with an excellent agreement between the results
from the 2D and 3D models in terms of magnitude and characteristic frequency of attenuation.
Indeed this was expected due to the radial symmetry of the spherical interfacial water film.
This outcome indicates that 3D effects are small for the adopted geometry. Furthermore,, the
results based on simple 2D models approximate well according to the dissipation magnitude
and frequency dependence of their corresponding 3D scenarios. The difference in the overall
value of the real-valued Young's modulus is associated with a larger relative quantity of soft
GH and a lower relative quantity of stiff quartz in the 3D model.



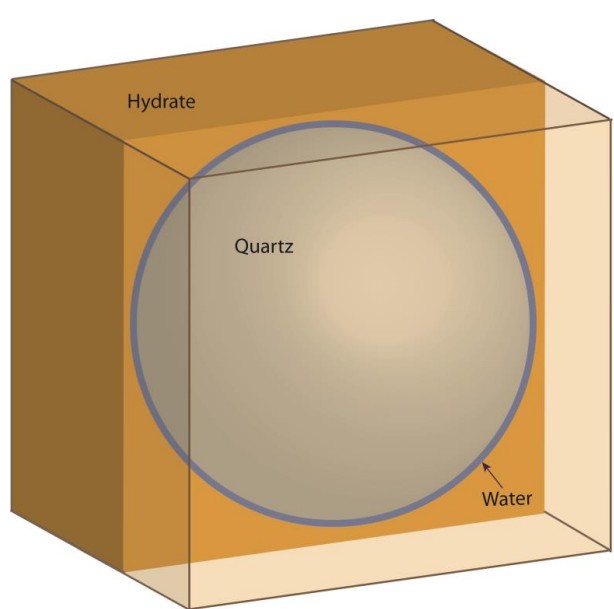

2  **Figure 14**: The 3D counterpart of the model shown in Figure 7: Fundamental block of a periodic
3  medium representing unconsolidated quartz grains which are separated from the embedding GH
4  background by a thin interfacial water film.





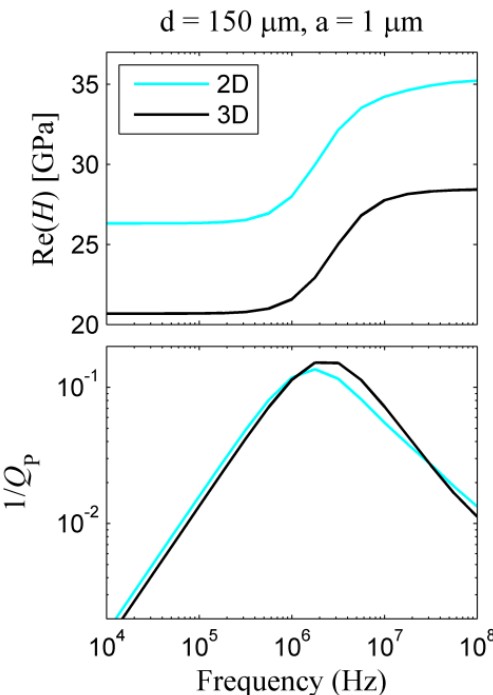

**Figure 15.** Real part of P-wave modulus, *H*, and corresponding P-wave attenuation, $1/Q_P$, as functions
of frequency, for the 2D model shown in Figure 7 and for its 3D counterpart shown in Figure 14. The
grain diameter d and thickness a of the interfacial water film are indicated in the plot title. The fields
shown in Figures 9-11 correspond to this 2D simulation.
**5. CONCLUSIONS**
Thin interfacial water films between sediment grains and the embedding GH matrix have
recently been observed in GH-bearing sediments through synchrotron-based micro-
tomography at a spatial resolution down to 0.38 μm. Based on these data, the appearance and
thicknesses of thin interfacial water films have been (geometrically) determined. With this
knowledge a new conceptual squirt flow model which refers to a spherical thin fluid film
coating the solid grains was introduced. The novelty of this model is constituted with respect
to its geometry, as compared to classical squirt flow models that involved interconnected
microcracks or microcracks connected to spherical pores instead of interfacial fluid films.
Numerical simulations were performed to calculate the energy dissipation in the proposed
model, considering a range of scenarios. Our results show that squirt flow in thin interfacial
water films can cause large and frequency-dependent attenuation in a broad frequency range
including seismic frequencies. Additionally, this effect does depend upon the interfacial water
films being connected to any other type of pore.
A numerical solution based on a set of coupled equations that reduce to Hooke's law in the
subdomains of the model corresponding to the elastic solid materials (grains and GH) and to



the quasi-static, linearized Navier-Stokes equations in the subdomains corresponding to the fluid (water) has been used. The results for our conceptual model show that the attenuation peak is shifted to lower frequencies with decreasing thickness of the interfacial water film and with increasing grain size (or the length of the film), as analogously known for microcrack aperture and length in classical squirt flow models. Furthermore, we tested the effect of inserting water pockets in an embedding GH matrix and the effect of connecting two neighboring thin interfacial water films through a water bridge. In general, the water bridges have a stronger effect on energy dissipation than the water pockets. Introducing such connections between neighboring interfacial water films causes a broadening of the attenuation spectrum towards higher frequencies. On the other hand, the presence of water pockets in the GH background only causes a slight overall increase in attenuation. Although the majority of our simulations were performed for 2D models, additional results of a 3D simulation showed that 3D effects are small for the basic 2D models that we have considered.

Our results represent a strong base to explain fundamental processes in GH bearing sediments and support previous speculations (Guerin and Goldberg, 2002; Dvorkin and Uden, 2004, Priest et al., 2006) that squirt flow is an important attenuation mechanism in GH-bearing sediments, even at frequencies as low as those in the seismic range. This strengthens the perception that P-wave attenuation may be used as an indirect geophysical attribute to estimate GH saturation. Nevertheless, further studies considering more realistic geometries for the microstructure of GH bearing sediments are necessary for a successful strategy to estimate GH saturations where hydrate is distributed in a dispersed manner instead of massive layers. For such a following study, our aim is to implement the segmented 3D images obtained from synchrotron-based micro-tomography as a direct model input for numerical investigations. At the moment this approach is challenging due to the corresponding large computational demand and requires additional segmentation steps for the 3D images, such as to allow for a smoothing of the stairs-like resolution artifacts at the boundaries of the interfacial water films. Furthermore, the image segmentation bears significant errors concerning the accuracy of the film thickness.

**ACKNOWLEDGEMENTS**

The authors thank the staff of the GZG crystallography group headed by Prof. W.F. Kuhs of the Georg August University Göttingen for their collaboration during the in-situ experiments at the TOMCAT beamline (Paul Scherrer Institute in Villigen, Switzerland) in 2012 and 2013. The presented work was co-funded by the German Science Foundation (DFG grant Ke 508/20 and Ku 920/18).

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

A Counter-Current Heat-Exchange Reactor for the Thermal Stimulation of Hydrate-Bearing
Sediments, Energies, 6(6), 3002-3016.
Sell, K., Saenger, E.H., Falenty, A., Chaouachi, M., Haberthür, D., Enzmann, F., Kuhs, W.F.,
Kersten, M. (2016), On the path to the digital rock physics of gas hydrate-bearing sediments –
processing of in situ synchrotron-tomography data, 7, p. 1243-1258,
Sloan, E. D., and C. A. Koh (2008), Clathrate hydrates of natural gases, Third edition, CRC
Press, Boca Raton, USA.
Spangenberg, E., and Kulenkampff, J. (2006), Influence of methane hydrate content on
electrical sediment properties, Geophys. Res. Lett., 33(24), 5.
Spangenberg, E. P., M.; Heeschen, K.; Schicks, J. M. (2015), Are Laboratory-Formed Hydrate-
Bearing Systems Analogous to Those in Nature?, J. Chem. Eng. Data 60(2), 258-268.
Subramaniyan, S., B. Quintal, C. Madonna, and E. H. Saenger (2015), Laboratory-based
seismic attenuation in Fontainebleau sandstone: Evidence of squirt flow, Journal of
Geophysical Research – Solid Earth, 120, 7526-7535, doi: 10.1002/2015JB012290.
Tisato, N., and B. Quintal (2013), Measurements of seismic attenuation and transient fluid
pressure in partially saturated Berea sandstone: evidence of fluid flow on the mesoscopic scale:
Geophysical Journal International, 195(1), 342-351, doi: 10.1093/gji/ggt259.
Tisato, N., and B. Quintal (2014), Laboratory measurements of seismic attenuation in
sandstone: Strain versus fluid saturation effects, Geophysics, 79, WB9-WB14, doi:
10.1190/geo2013-0419.1.
Waite, W. F., et al. (2009), Physical properties of hydrate-bearing sediments, Rev. Geophys.,
32  47, 38.
White, J. E., 1975, Computed seismic speeds and attenuation in rocks with partial gas
saturation: Geophysics, 40, 224-232, doi: 10.1190/1.1440520.
White, J. E., N. G. Mikhaylova, and F. M. Lyakhovitskiy, 1975, Low-frequency seismic waves
in fluid- saturated layered rocks: Izvestiya, Academy of Sciences, USSR. Physics of the Solid
Earth, 11, 654-659.
Winkler, K. W., and A. Nur (1982). Seismic attenuation: Effects of pore fluids and frictional-
sliding. Geophysics, 47, 1-15, doi: 10.1190/1.1441276.
Yun, T. S., F. M. Francisca, J. C. Santamarina, and C. Ruppel (2005), Compressional and shear
wave velocities in uncemented sediment containing gas hydrate, Geophys. Res. Lett., 32(10),
44  5.



1    Zhang, Q., F. G. Li, C. Y. Sun, Q. P. Li, X. Y. Wu, B. Liu, and G. J. Chen (2011),
2    Compressional wave velocity measurements through sandy sediments containing methane
3    hydrate, Am. Miner., 96(10), 1425-1432.
