# Peer review of "Squirt flow due to interfacial water films in hydrate bearing sediments"

_Solid Earth, 2017_

## Referee Comment (RC1) · Anonymous Referee #1 · 23 Oct 2017

The paper presents a model study on absorption based on a squirt flow model in hydrate-bearing sediments. The setup of the model is straight forward and based on visual observations of thin (sub-micron) water films between quartz sand grains and clathrate. The mechanism which creates a pressure gradient and following flow in the water film is described clearly and also the influences of different water film thickness, different grain sizes, presence of isolated water pockets in the hydrate and, the influence of connections between the water films. The shift of the maximum in the dependence of $1/Q$ on frequency with changing thickness of the water film shows, that a distribution of various film thicknesses would result in high absorption ($1/Q$) over a broad frequency range. This is what one would expect, because the high absorption of hydrate-bearing sediments has been observed in the field at seismic frequencies and in

the lab at ultrasonic frequencies. The paper provides a valuable contribution towards the understanding of possible absorption mechanisms in hydrate-bearing sediments and should be published soon. However, to avoid the "misuse" of the model in the interpretation of real measurements the author should clearly state what the restrictions and limits of the model are. The visual observations used for the modelling should also brought in relation to other visual observations (see comment/reference below).

The following two main restrictions, at least, should be pointed out to the reader: The model is based on the observations/results from high-resolution synchrotron-based X-ray micro- tomography, where the hydrate is produced with the "gas in excess method". The method used for the hydrate formation is essential to understand the resulting hydrate habit. The "gas in excess method" forms a grain coating hydrate structure (with a water film between hydrate and grains), because the water which is wedding the grains is transformed into hydrate. When hydrate is formed with the "water in excess method" the grains will also be water wet, but these very thin (sub-micron) hydrate films between the grains and the hydrate structure will only occur at very high hydrate saturations (the highest reported values to my knowledge are about 90% from Mallik and the Gulf of Mexico ).

See also Tohidi's paper: "Gas bubbles, when present, act as preferential nucleation sites, but silica glass surfaces are wetted strongly by water and do not promote heterogeneous surface nucleation; a surface water film remains to high clathrate saturations. The fact that hydrates grow within the center of pores, rather than on grain surfaces, is likely to restrict the potential for cementation of sediments, unless a large proportion of the pore space is filled with hydrate."

Tohidi, B., Anderson, R., Clennell, M. B., Burgass, R. W., & Biderkab, A. B. (2001). Visual observation of gas-hydrate formation and dissociation in synthetic porous media by means of glass micromodels. Geology, 29(9), 867-870.

1) This model with sub-micron bound-water films is restricted to very high hydrate saturations (for your model with 250 – 150 $\mu$m grain size and a water film below 1 $\mu$m I calculated about 99% hydrate saturation) or to gas-bearing reservoirs where the free water, available for hydrate formation, has been completely transformed into hydrate.

The model (e.g. Fig. 7 & Fig. 12) assumes the sand grain as an inclusion in the hydrate matrix (a suspension of quartz grains in hydrate). This neglects the fact that hydrate is a secondary phase forming in the pore space when the sediment already has deposited and forms a grain skeleton with grain-to-grain contacts. Depending on the number and size of these contacts (compaction, overburden) the modulus (mainly the real part of the complex modulus) of the hydrate free grain skeleton will vary. Q is derived from the ratio of imaginary part and the real part of the complex modulus and will, therefore, change when the real part changes due to different number of grain-to-grain contact (coordination number).

2) The specific properties of the sediment grain skeleton and the resulting influence on absorption are not considered.

To study this special squirt-flow mechanism related to the existence of thin water films initially separated from other influences is certainly justified. However, this model can be improved in future to also involve effects from the grain skeleton (e.g. involving Hertz-Mindlin theory) and it can be combined with other absorption mechanisms (see Marin-Moreno's paper).

---

## Referee Comment (RC2) · Anonymous Referee #2 · 20 Nov 2017

Dear authors,

I found your paper intriguing and comprehensive; in my understanding, you provide previously published observational evidence from x-ray tomography to support the claim that a thin water film around sand grains embedded in a gas hydrate matrix is a good conceptual model that captures the high attenuation observed in gas hydrate systems.

I believe that the general scope of your paper deserves some attention as squirt flow in hydrates is only recently being considered as the responsible mechanism and Marin-Moreno et al. (2017) is potentially too confusing for scientists to use as it considers the overlap of many mechanisms. So there is definitely a gap in the literature for simple, usable models of the squirt flow of GH and I think your paper is a step towards the right direction. I do however think that the presentation of your work does not do the ideas

justice and as a result lessens the potential significance it may have. Below are some of my most serious concerns:

1. I am not entirely familiar with imaging techniques when applied to hydrates so I am not aware how the conceptualisation of your model is affected by the imaging. I realise the experimental imaging results are presented elsewhere but I would still like to see a convincing argument about how the thin water film surrounding a quartz grain within a hydrate is indeed a physically plausible configuration rather than an imaging artifact

2. Your single circular grain model presented in Figure 7 is the exact same model proposed by White, J. (1975) which you cite in passing in your introduction. The only difference here is that your sand grain is in place of a second fluid in White's model. This is nowhere mentioned and I firmly believe it should be

3. You claim to numerically solve (1), (2) but you show no meshing and mention no restrictions on your domains (is the circular sand grain obeying a free BC, is it fixed etc?)

4. As I mentioned earlier in comment 2 this model is exactly the same as White's model which has an exact analytic solution. Why does your model of figures 7,14 not have an analytic solution despite the simple domain and, if it does, why are we not seeing it - it is so much easier for someone to replicate your work if they have a formula to use. Does your model agree with White's model if his second fluid becomes really stiff (to the limit of a sand grain)?

5. Although these may be commonplace for people familiar with squirt flow, how do you define "mesoscopic" as a scale here? What are the domains and boundary conditions that go into solving your equations? How does the relative rather than absolute scaling affect the behaviour of your attenuation curves ? What I mean here is that if you fixed the GH square in model 7 to have side = 1 you could see the affect of relative saturation of GH and water rather than inserting absolute values. This would be much more illuminating than your figure 8. This problem is also present when you discuss

water bridges and your model demonstrates a second peak in the attenuation curves but the reader is left wondering how(if?) does this peak move when the bridge gets longer. There is significant mathematical rigour that is missing from your work which is not in itself always a bad thing but this impedes the impact and significance it may have.

6. You mention shear dispersion in passing indicating that you have numerically calculated it ("it can be calculated in a similar manner simply by changing the boundary conditions") - is the shear dispersion predicted by this model in any way realistic? I feel that it would be beneficial for your work to show the attenuation and dispersion of shear velocity and discuss the success/limitation of your modelling strategy with respect to shear.

And some more minor comments:

-Figure 2 have some labels GH* and I have not been able to see what the * refers to -Figure 3 caption has an unrendered mu character that shows up as a box -P20L5 needs a space between "effect" and "of"

---

## Author Comment (AC1) · 16 Dec 2017

Squirt flow due to interfacial water films in hydrate bearing sediments

Kathleen Sell[1], Beatriz Quintal[2], Michael Kersten[1] and Erik H. Saenger[3,4]

[1]Johannes Gutenberg-University Mainz, Germany

[2]University of Lausanne, Switzerland

[3] International Geothermal Centre, University of Applied Sciences Bochum, Germany

[4] Ruhr University Bochum, Germany

*Correspondence to:* Kathleen Sell (sell@uni-mainz.de)

*Please note: All our responses to remarks of reviewers are in red and italic.*

*Dear Anonymous Referee #1,*

*We appreciate the time, interest and effort you invested to evaluate our manuscript. In the following, we respond to your questions, comments and concerns in order of appearance, to improve our manuscript based on your valued input.*

*Kind Regards,*

*Kathleen Sell, Beatriz Quintal, Michael Kersten, and Erik H. Saenger*
* * *
Anonymous Referee #1

Review type: Interactive comment

https://doi.org/10.5194/se-2017-106, 2017

The paper presents a model study on absorption based on a squirt flow model in hydrate-bearing sediments. The setup of the model is straight forward and based on visual observations of thin (sub-micron) water films between quartz sand grains and clathrate. The mechanism which creates a pressure gradient and following flow in the water film is described clearly and also the influences of different water film thickness, different grain sizes, presence of isolated water pockets in the hydrate and, the influence of connections between the water films. The shift of the maximum in the dependence of 1/Q on frequency with changing thickness of the water film shows, that a distribution of various film thicknesses would result in high absorption (1/Q) over abroad frequency range. This is what one would expect, because the high absorption of hydratebearing sediments has been observed in the field at seismic frequencies and in the lab at ultrasonic frequencies. The paper provides a valuable contribution towards the understanding of possible absorption mechanisms in hydrate-bearing sediments and should be published soon.

However, to avoid the "misuse" of the model in the interpretation of real measurements the author should clearly state what the restrictions and limits of the model are. The visual observations used for the modelling should also brought in relation to other visual observations (see comment/reference below). The following two main restrictions, at least, should be pointed out to the reader:

The model is based on the observations/results from high-resolution synchrotron-based Xray micro- tomography, where the hydrate is produced with the "gas in excess method". The method used for the hydrate formation is essential to understand the resulting hydrate habit. The "gas in excess method" forms a grain coating hydrate structure (with a water film between hydrate and grains), because the water which is wedding the grains is transformed into hydrate. When hydrate is formed with the "water in excess method" the grains will also be water wet, but these very thin (sub-micron) hydrate films between the grains and the hydrate structure will only occur at very high hydrate saturations (the highest reported values to my knowledge are about 90% from Mallik and the Gulf of Mexico ).

*Authors: As suggested by the reviewer we added the mandatory information in the Introduction as well as in section 2.*

See also Tohidi's paper: "Gas bubbles, when present, act as preferential nucleation sites, but silica glass surfaces are wetted strongly by water and do not promote heterogeneous surface nucleation; a surface water film remains to high clathrate saturations. The fact that hydrates grow within the center of pores, rather than on grain surfaces, is likely to restrict the potential for cementation of sediments, unless a large proportion of the pore space is filled with hydrate."

Tohidi, B., Anderson, R., Clennell, M. B., Burgass, R. W., & Biderkab, A. B. (2001). Visual observation of gas-hydrate formation and dissociation in synthetic porous media by means of glass micromodels. Geology, 29(9), 867-870.1)

This model with sub-micron bound-water films is restricted to very high hydrate saturations (for your model with 250 – 150 m grain size and a water film below 1µm calculated about 99% hydrate saturation) or to gas-bearing reservoirs where the free water, available for hydrate formation, has been completely transformed into hydrate.

*Authors: Indeed, the information that for our type of model the assumed GH saturation will be very high <90% was missing. Therefore, this fact has been added to the Introduction section.*

The model (e.g. Fig. 7 & Fig. 12) assumes the sand grain as an inclusion in the hydrate matrix (a suspension of quartz grains in hydrate). This neglects the fact that hydrate is a secondary phase forming in the pore space when the sediment already has deposited and forms a grain skeleton with grain-to-grain contacts. Depending on the number and size of these contacts (compaction, overburden) the modulus (mainly the real part of the complex modulus) of the hydrate free grain skeleton will vary. Q is derived from the ratio of imaginary part and the real part of the complex modulus and will, therefore, change when the real part changes due to different number of grain-to-grain contact (coordination number). 2) The specific properties of the sediment grain skeleton and the resulting influence on absorption are not considered.

*Authors: It is true that our model is a very simplified approach regarding sedimentary systems with respect to grain contacts and therefore a first step towards more realistic matrices as stated in the conclusion part. We are aiming for SRXCT/HRXCT data input to extend our model approach. But for now we are limited to the simple scenario of unconsolidated sediments.*

*We added your valuable comment to our Results section.*

To study this special squirt-flow mechanism related to the existence of thin water films initially separated from other influences is certainly justified. However, this model can be improved in future to also involve effects from the grain skeleton (e.g. involving Hertz-Mindlin theory) and it can be combined with other absorption mechanisms (see Marin-Moreno's paper).

*Authors: Further investigations involve the stepwise extension of this model towards more realistic settings is aimed but hampered by the lack of a segmentation routine capable to cover a full dataset (24GB). Currently a machine learning code is tested on the data to handle this issue.*

---

## Author Comment (AC2) · 16 Dec 2017

Squirt flow due to interfacial water films in hydrate bearing sediments

Kathleen Sell[1], Beatriz Quintal[2], Michael Kersten[1] and Erik H. Saenger[3,4]

[1]Johannes Gutenberg-University Mainz, Germany

[2]University of Lausanne, Switzerland

[3] International Geothermal Centre, University of Applied Sciences Bochum, Germany

[4] Ruhr University Bochum, Germany

*Correspondence to:* Kathleen Sell ([sell@uni-mainz.de](mailto:sell@uni-mainz.de))

*Please note: All our responses to remarks of reviewers are in red and italic.*

*Dear Anonymous Referee #2,*

*We appreciate the time, interest and effort you invested to evaluate our manuscript. In the*
*following, we respond to your questions, comments and concerns in order of appearance, to*
*improve our manuscript based on your valued input.*

*Kind Regards,*

*Kathleen Sell, Beatriz Quintal, Michael Kersten, and Erik H. Saenger*

* * *

Anonymous Referee #2

Review type: Interactive comment

https://doi.org/10.5194/se-2017-106, 2017

Dear authors,

I found your paper intriguing and comprehensive; in my understanding, you provide previously
published observational evidence from x-ray tomography to support the claim that a thin water
film around sand grains embedded in a gas hydrate matrix is a good conceptual model that
captures the high attenuation observed in gas hydrate systems. I believe that the general scope
of your paper deserves some attention as squirt flow in hydrates is only recently being
considered as the responsible mechanism and Marin-Moreno et al. (2017) is potentially too confusing for scientists to use as it considers the overlap of many mechanisms. So there is
definitely a gap in the literature for simple, usable models of the squirt flow of GH and I think
your paper is a step towards the right direction. I do however think that the presentation of your
work does not do the ideas justice and as a result lessens the potential significance it may have.
Below are some of my most serious concerns:

1. I am not entirely familiar with imaging techniques when applied to hydrates so I am not
aware how the conceptualisation of your model is affected by the imaging. I realise the
experimental imaging results are presented elsewhere but I would still like to see a convincing
argument about how the thin water film surrounding a quartz grain within a hydrate is indeed a
physically plausible configuration rather than an imaging artifact

*Authors: A common image artifact occurring when conducting synchrotron-based tomography*
*is the so-called edge enhancement. Probably, this is the artifact you have in mind. When plotting*
*a histogram over an area where possible edge enhancement occurs the histogram line plot will*
*reveal symmetrical valleys and peaks. Here, this is not the case because we can identify a*
*several voxel wide interface between the GH and quartz. This interface is in the same gray-*
*value range than the water phase identified in the intial (untreated) samples – these samples*
*are completely GH free and we can be sure that the phase identified is water. The observation*
*of the interfacial water layer from the experimental results of Chaouachi et al. (2015) is in*
*accordance with the publication of Tohidi et al. (2001). Additionally several molecular*
*numerical simulations showed that a water layer prefers the interface of GH and quartz grains*
*(Bagherzadeh et al., 2012; Bai et al., 2011; Liang et al., 2011). For the matter of clarification*
*text passages have been added to the manuscript.*

2. Your single circular grain model presented in Figure 7 is the exact same model proposed by
White, J. (1975) which you cite in passing in your introduction. The only difference here is that
your sand grain is in place of a second fluid in White's model. This is nowhere mentioned and
I firmly believe it should be.

*Authors: Our model might, in principle, resemble White's model from the spherical geometries*
*involved, but it is considerably different. White's model refers to a spherical porous patch*
*embedded in a porous background. Fluid pressure diffusion occurs between those two*
*poroelastic subdomains across the spherical surface. The model that we consider refers to a*
*non-porous solid spherical inclusion separated from the embedding non-porous solid*
*background by a thin liquid shell. In this case, fluid pressure diffusion occurs only within the*
*liquid shell, tangentially to its spherical surfaces.*

3. You claim to numerically solve (1), (2) but you show no meshing and mention no restrictions
on your domains (is the circular sand grain obeying a free BC, is it fixed etc?)

*Authors: We have added a figure with a mesh for the main model (new Figure 8) and all the*
*necessary BC are explained in the Numerical Methodology section.*

4. As I mentioned earlier in comment 2 this model is exactly the same as White's model which
has an exact analytic solution. Why does your model of figures 7,14 not have an analytic
solution despite the simple domain and, if it does, why are we not seeing it - it is so much easier
for someone to replicate your work if they have a formula to use. Does your model agree with
White's model if his second fluid becomes really stiff (to the limit of a sand grain)?

*Authors: Our model is different than White's model, as explained above. We believe this is*
*clearer after our revision.*

5. Although these may be commonplace for people familiar with squirt flow, how do you define
"mesoscopic" as a scale here? What are the domains and boundary conditions that go into
solving your equations? How does the relative rather than absolute scaling affect the behaviour
of your attenuation curves? What I mean here is that if you fixed the GH square in model 7 to
have side = 1 you could see the affect of relative saturation of GH and water rather than inserting
absolute values. This would be much more illuminating than your figure 8. This problem is also
present when you discuss water bridges and your model demonstrates a second peak in the
attenuation curves but the reader is left wondering how(if?) does this peak move when the
bridge gets longer. There is significant mathematical rigour that is missing from your work
which is not in itself always a bad thing but this impedes the impact and significance it may
have.

*Authors: Our model is not at the mesoscopic scale, but microscopic. With respect to*
*mathematical rigor, we believe that we gave the necessary information, such as the equations,*
*the parameter values, the model geometry, and the boundary conditions are described in the*
*numerical methodology part.*

6. You mention shear dispersion in passing indicating that you have numerically calculated it
("it can be calculated in a similar manner simply by changing the boundary conditions") - is the
shear dispersion predicted by this model in any way realistic? I feel that it would be beneficial
for your work to show the attenuation and dispersion of shear velocity and discuss the
success/limitation of your modelling strategy with respect to shear.

*Authors: Unfortunately our code becomes unstable under the boundary condition necessary for*
*a shear test and the results for S-wave attenuation and dispersion at this point are not reliable.*
*The compressional tests to obtain P-wave attenuation and dispersion, on the other hand, have*
*been tested through comparisons with other solutions (e.g., Quintal et al, 2016, Geophysics)*
*and yield stable and reliable results.*

And some more minor comments:

- Figure 2 have some labels GH* and I have not been able to see what the * refers

- Figure 3 caption has an unrendered mu character that shows up as a box

- P20L5 needs a space between "effect" and "of"

*Authors: These mistakes have been fixed.*